# TIM-3 Is a Potential Immune Checkpoint Target in Cats with Mammary Carcinoma

**DOI:** 10.3390/cancers15020384

**Published:** 2023-01-06

**Authors:** Sofia Valente, Catarina Nascimento, Andreia Gameiro, João Ferreira, Jorge Correia, Fernando Ferreira

**Affiliations:** 1CIISA—Center of Interdisciplinary Research in Animal Health, Faculty of Veterinary Medicine, University of Lisbon, Avenida da Universidade Técnica, 1300-477 Lisboa, Portugal; 2Associate Laboratory for Animal and Veterinary Sciences (AL4AnimalS), 1300-477 Lisbon, Portugal; 3iMM João Lobo Antunes, University of Lisbon, Av. Prof. Egas Moniz, 1649-028 Lisbon, Portugal

**Keywords:** feline mammary carcinoma, TIM-3, tumor-infiltrating lymphocytes, tumor microenvironment, immunotherapy

## Abstract

**Simple Summary:**

Feline mammary carcinomas are highly prevalent tumors, with aggressive behavior and scarce therapeutic options. Thus, there is a high need for novel biomarkers and therapeutic targets. In human medicine, immune checkpoint inhibitors targeting TIM-3 are emerging as promising treatment strategies; however, the clinical relevance of immune checkpoints in feline cancers is sparse. To unravel the clinical importance of TIM-3 in feline mammary carcinoma, we analyzed the intratumor expression and serum levels of the TIM-3 receptor in tumor-bearing cats. The results obtained show that TIM-3 is highly expressed in both immune and cancer cells, and the localization of TIM-3 expression can predict clinical outcomes in feline mammary carcinoma. TIM-3 levels in serum were reduced in diseased compared to healthy animals. These findings shed light on the biological role of TIM-3 in feline mammary carcinoma and support clinical testing of novel therapies targeting TIM-3.

**Abstract:**

Recent findings in human breast cancer (HBC) indicate that T-cell immunoglobulin and mucin-domain-containing molecule-3 (TIM-3)-targeted therapies may effectively activate anticancer immune responses. Although feline mammary carcinoma (FMC) is a valuable cancer model, no studies on TIM-3 have been developed in this species. Thus, we evaluated the expression of TIM-3 by immunohistochemistry in total (t), stromal (s), and intra-tumoral (i) tumor-infiltrating lymphocytes (TILs) and in cancer cells, of 48 cats with mammary carcinoma. In parallel, serum TIM-3 levels were quantified using ELISA and the presence of somatic mutations in the TIM-3 gene was evaluated in 19 tumor samples. sTILs-TIM3+ were more frequent than iTILs-TIM-3+, with the TIM-3 ex-pression in sTILs and cancer cells being associated with more aggressive clinicopathological features. In contrast, the TIM-3 expression in iTILs and tTILs was associated with a more benign clinical course. Moreover, the serum TIM-3 levels were lower in animals with FMC when compared to healthy animals (*p* < 0.001). Only one somatic mutation was found in the TIM-3 gene, at intron 2, in one tumor sample. Altogether, our results suggest that the expression of TIM-3 among TILs subpopulations and cancer cells may influence the clinical outcome of cats with FMC, in line with the previous reports in HBC.

## 1. Introduction

Similarly to women, female cats are immunocompetent animals in which aggressive mammary carcinomas represent a major proportion of cancer-related deaths [1,2,3]. Additionally, cats are exposed to a similar environment as humans, exhibiting an accurate representation of the interaction between genetics and environmental risk factors [4]. Feline mammary carcinoma (FMC) represents a reliable cancer model for the study of human breast cancer (HBC). In fact, both tumors occur spontaneously [4], have a high homology within cancer-related genes [5], and share similar metastatic patterns to regional lymph nodes, lungs, liver, and pleura [6,7]. Moreover, FMC shares many clinicopathological, epidemiological, histological, and molecular features with HBC [8]. In general, FMC has a high mortality rate (80–90%) [2] because therapeutic options are scarce, and mostly consisting of uni/bilateral total or partial mastectomy, with or without chemotherapeutic adjuvant protocols [9]. Hence, there is a high need for novel effective therapeutic options in FMC. For breast cancer patients, immune checkpoint inhibitors are emerging as promising treatment options, mostly through the blockade of molecules that inhibit the activation of exhausted T-cells [10]. Although these therapeutic targets have improved treatment outcomes for several advanced human cancers [11,12], their efficacy is limited to a proportion of patients [13] and side effects are common (e.g. severe autoimmune-like reactions), especially when anti-PD-1 are combined with anti-CTLA-4 [14]. In this context, new inhibitory checkpoint molecules are being investigated as an attempt to surpass the drawbacks of immune checkpoint inhibitors (ICIs). Recently, T-cell immunoglobulin and mucin-domain-containing molecule-3 (TIM-3) has emerged as a promising target for cancer immunotherapy. In fact, resistance to current ICIs can be associated with a compensatory upregulation of other immune checkpoints, such as TIM-3 [15]. In HBC, this novel ICI has been demonstrated to play a crucial part in autoimmune diseases, chronic viral infections, and tumors, with remarkable results in treatment outcome and prognosis [16]. The TIM-3 molecular structure includes four parts: the immunoglobulin (Ig) V domain, the mucin domain, the transmembrane region, and the cytoplasmic region [17]. To date, four relevant ligands have been shown to interact with TIM-3 by binding to different regions on the extracellular IgV domain and generating distinct effects upon binding. These ligands include galectin-9 (Gal-9), high-mobility group protein B1 (HMGB1), carcinoembryonic antigen cell adhesion molecule 1 (Ceacam-1), and phosphatidylserine (PtdSer) [17]. For T cells, a Bat-3-mediated regulation of TIM-3 has been described. Indeed, Bat-3 binds to the cytoplasmic tail of TIM-3 and inhibits its function, by controlling signaling pathways that affect the T-cell receptor [18]. When TIM-3 binds to either Gal-9 or Ceacam-1 [19,20], the conserved tyrosine residues (Y256 and Y263) in the cytoplasmic tail undergo phosphorylation, resulting in the detachment of Bat-3, thus allowing TIM-3 to play its inhibitory function by inactivation of Lck and, consequently, downregulation of the TCR proximal signaling [18]. Particularly in breast cancer cells, the activation of the NF-κB/IL-6/STAT3 pathway has been associated with a pro-tumoral response, by regulating downstream genes that control cell proliferation and angiogenesis [21]. Considering its inhibitory function on both TILs and cancer cells, evidence shows that targeting TIM-3 might be a promising treatment option in solid tumors. In fact, several experimental trials are investigating multiple anti-TIM-3 mAbs, in monotherapy or in combination with other ICI or chemotherapy [22].

In humans, several reports have demonstrated the importance of TIM-3 expression on TILs and cancer cells in a variety of tumors, including HBC [23,24,25,26,27,28,29]. Indeed, the expression of TIM-3 has been correlated with a worse prognosis among several tumor types [30]. Nevertheless, in breast cancer, the prognostic role of TIM-3 seems to differ among the breast cancer subtypes. Accordingly, a higher TIM-3 expression on the luminal carcinoma subtypes (luminal A and luminal B) has been associated with a worse prognosis, whereas in basal-like triple-negative breast cancer, increased TIM-3 levels were associated with a more favorable prognosis [16]. Thus, in this study, our main goals were to evaluate the TIM-3 expression in TILs with different tumor-related locations (stromal and intra-tumoral) and in cancer cells, to compare the serum TIM-3 levels of cats with mammary carcinoma with those of healthy controls, and to identify putative mutations in the feline TIM-3 gene. In addition, we investigated whether associations with clinicopathological features and other important immune biomarkers were present. 

## 2. Materials and Methods

### 2.1. Study Population

This study included 48 queens, previously diagnosed with mammary carcinoma, whose medical records (Table 1) were provided by the Veterinary Hospital of the Faculty of Veterinary Medicine (University of Lisbon), where the animals were diagnosed and treated. All animals enrolled in the study were subjected to a mastectomy, allowing for a representative sampling of the mammary tumor tissue. These tumor samples were immediately placed in a 10% formalin solution for 24–48 h, and then were dehydrated through a series of graded alcohol baths and immersed in paraffin wax. Additionally, blood samples, from the diseased queens and from 14 healthy cats used as controls, were collected for the serum TIM-3 evaluation. Subsequently, the blood samples were centrifugated and stored at −80 °C.

### 2.2. Immunohistochemical Staining and Analysis

To evaluate the TIM-3 expression on the tumor tissue samples, the feline mammary carcinoma formalin-fixed paraffin-embedded (FFPE) blocks were cut into 3 μm sections using a microtome. The tissue sections were then placed on adhesive glass slides, which were incubated at 64 °C for 60 min and then at 37 °C overnight. 

Regarding the immunohistochemistry (IHC) technique, firstly, the slices underwent a pre-treatment process, conducted on PT-Link (Dako, Agilent, Santa Clara, CA, USA). Antigen retrieval was performed for 20 min at 96 °C, using retrieval solution at pH 6.0, the citrate buffer (Dako, K800521-2). Afterwards, the slides cooled down for 30 min at room temperature (RT) and were washed with distilled water (2 × 5 min).

The IHC technique was performed with the solutions from the Novolink Polymer Detection System Kit (RE7280-CE, Leica Biosystems, Newcastle, UK). Before antibody incubation, tissue samples were treated for the endogenous peroxidase activity, with a peroxidase block for 15 min. Then, the nonspecific antigenic binding was prevented by incubating the tissue slices with the Protein Block solution for 10 min. Finally, the tissue samples were incubated at RT for 1 h, in a humidified chamber, with a rabbit anti-TIM-3 polyclonal antibody (ab185703, Abcam, Cambridge, UK), diluted at 1:300. All protocol procedures were followed by two washing steps with Phosphate-buffered saline (PBS) for 5 min each. Further, the detection was performed using the Novolink Polymer, incubated for 30 min, at RT. Following an additional washing step (two PBS washes for 5 min each), the staining was performed, and each tissue section was incubated for 5 min with DAB chromogen (diaminobenzidine) diluted at 1:20 in Novolink DAB substrate buffer. Finally, after being counterstained with Harris hematoxylin for 5 min, tissue sections were dehydrated in an ethanol gradient and xylene and covered with a thin glass coverslip.

For the evaluation of tissue sections, human tonsil and feline lymph node tissues served as positive controls, whereas feline mammary tissues were used as negative controls. The percentage of cells expressing TIM-3 and the staining intensity were evaluated for each tumor tissue section, from five individual fields at high resolution (400× magnification). Then, the average of stained immune and cancer cells, in the five evaluated fields, was calculated. The five fields that underwent examination were randomly selected, with necrotic areas and technical artifacts being discarded. Regarding the TILs, they were evaluated according to their tumor location, as follows: when in contact with tumor cells or within the tumor cell nests were defined as intra-tumoral, whereas when distributed in the interstitial stroma surrounding the cancer cells were defined as stromal. For TILs (stromal and intra-tumoral) and tumor cells, TIM-3 positivity was defined as staining > 1%. The scores of percentages of TILs-TIM-3+ were recorded as: 0 (<1%), 1 (1–10%), 2 (11–24%), 3 (25–50%), and 4 (≥50%). For cancer cells-TIM-3+, the percentage scores were as follows: 0 (<1%), 1 (1–24%), 2 (25–90%), and 3 (≥90%). 

### 2.3. Measurement of Serum TIM-3

The serum TIM-3 levels were measured by an ELISA technique and using a commercial kit (Human TIM-3 Quantikine ELISA Kit, DTIM30, R&D systems, Minneapolis, MN, USA). The kit components were stored at 4 °C. 

The ELISA technique was performed in a 96-well microplate coated with a monoclonal antibody specific for TIM-3. Then, serum samples were submitted to a 10-fold dilution in a buffered protein base solution (Calibrator Diluent, RD5-24), and 50 μL of standards and diluted serum samples, from the controls and diseased animals, were added to the wells. At this point, the plates were incubated for 1 h, at RT, on a horizontal microplate shaker set at 500 ± 50 rpm. After a washing step (4 × 400 μL/well per wash) with wash buffer, 200 μL of an enzyme-linked polyclonal antibody, specific for human TIM-3 and conjugated to horseradish peroxidase (HRP), was added to each well and the incubation step was repeated. Following an additional washing step with wash buffer (4 × 400 μL/well per wash), to remove the unbounded antibody-enzyme reagent, 200 μL of the substrate solution was added to each well and color developed in proportion to the amount of TIM-3 bound. Afterwards, the plate was incubated for 30 min, at RT, in the dark. Finally, the reaction was interrupted by adding 50 μL of stop solution (sulfuric acid) per well. 

The optical density was determined using a microplate reader set to 450 nm. To correct for optical imperfections in the plate, a second reading was performed at 570 nm and readings were subtracted from the readings at 450 nm.

Finally, the serum TIM-3 concentration (pg/mL) was determined using a four-parameter logistic (4-PL) curve-fit (r^2^ > 0.99). To build this graph, serial 2-fold dilutions of a stock solution of recombinant TIM-3 were performed (8000, 4000, 2000, 1000, 500, 250, 125, 62.5, 31.25, 15.63, 7.8, and 3.9 pg/mL).

### 2.4. DNA Amplification and Sequence Analysis of Feline TIM-3 Gene

The genomic DNA sequences used were extracted as previously described [31]. For the amplification of the regions of interest (exon 3) in the feline TIM-3 gene (NC_058368.1), one pair of specific primers was designed (Table 2) in the primer designing tool (NCBI). The PCR reaction mixture was obtained by adding the following reagents: Milli-Q water and DNA to maintain a final DNA concentration of 4 ng/mL, a standard reaction mixture (4.8 μL/sample), which resulted from the mix of 4 μL/sample of Phusion GC Buffer (F519L, Thermo Fischer Scientific, Waltham, MA, USA), 0.4 μL/sample of GRS dNTP Mix (GP010.0001, Grisp, Porto, Portugal), 0.1 μL/sample of the forward and reverse primers (Table 2), and 0.2 μL/sample of Phusion™ High-Fidelity DNA Polymerase (F-530XL, Thermo Fischer Scientific). The PCR technique was performed with a thermal cycler which was setup according to the following: denaturation at 98 °C for 30 s, followed by 35 cycles at 95 °C for 10 s, 52 °C for 30 s, 72 °C for 10 s, plus one extension step of 72 °C for 10 min. Then, a 2.5% agarose gel was prepared to evaluate whether the expected size for each amplified fragment was present. Afterwards, the 19 DNA fragments analyzed were purified and sequenced by the Sanger technique at StabVida (Caparica, Portugal).

The identification of the TIM-3 exon 3 was performed using the Basic Local Alignment Search Tool (NCBI), which compared the sequence of the feline TIM-3 gene (NC_058368.1) with the DNA samples submitted to evaluation, to find regions of similarity between the nucleotide sequences. Sequenced fragments were aligned using the ClustalW 2.1. tool (Bioedit software).

### 2.5. Statistical Analysis

Statistical analysis was conducted in IBM SPSS Statistics software version 28.0.1.0 (Armonk, NY, USA) and the two-sided *p*-value < 0.05 was considered statistically significant for a 95% confidence level (* *p* < 0.05 and ** *p* < 0.01). GraphPad Prism version 9.4.1 for Windows (GraphPad Software, La Jolla, CA, USA) was used to plot the graphs. The Kruskal–Wallis test and Dunn’s multiple comparisons post-test were applied to compare TIM-3 expression in TILs with different tumor locations (stromal, intra-tumoral, and total) and cancer cells between cats with different mammary carcinoma histological and molecular subtypes. The Mann–Whitney test was used to compare the TIM-3 expression levels in TILs and cancer cells with several clinicopathological features, to evaluate significant differences in TIM-3 serum levels between cats with mammary carcinoma and healthy controls and to assess the association between the TIM-3 serum levels and the TIM-3 expression on TILs and cancer cells. Correlations between the TIM-3 expression levels in TILs and cancer cells and different clinicopathological features were investigated using Spearman’s coefficient. Spearman’s coefficient was also used to assess the correlations between the serum TIM-3 levels and the previously measured serum levels of IL-6, TNF-α, CTLA-4, PD-L1, PD-1, VEGF-A, VEGFR-1, VEGFR-2, and LAG-3. 

Survival curves were plotted using the Kaplan–Meier method and the statistical significance between groups determined by the Log-rank test. Receiver operating characteristic (ROC) analysis and the Youden index were used to determine the optimal cut-point value for serum TIM-3 levels.

## 3. Results

### 3.1. TIM-3 Is Highly Expressed in TILs and in Cancer Cells in Feline Mammary Carcinoma, with sTILs-TIM-3+ Being More Frequent Than iTILs-TIM-3+

The immunohistochemistry analyses revealed that TIM-3 is highly expressed in TILs (intra-tumoral and stromal) (Figure 1a,c,d,e) and in cancer cells (Figure 1a,b,e,f), showing a nuclear-cytoplasmic pattern of staining, as previously reported [32]. Considering the tumor location of TILs, from the 45 tumor specimens evaluated, 93.3% showed sTILs-TIM-3+, whereas 64.4% of the tumors displayed iTILs-TIM-3+ (Table 3 and Table 4). Further analysis revealed a progressive decrease in the percentage of tumors having iTILs-TIM-3+ from the luminal A and luminal B subtypes to the HER-2+ and triple-negative carcinomas, as follows: luminal A subtype (83.3%), luminal B subtype (70.6%), HER-2+ subtype (66.7%), triple-negative normal-like (60.0%), and triple-negative basal-like subtypes (37.5%) (Table 3). In contrast, the percentage of tumors having sTILs-TIM-3+ was similar among the different mammary carcinoma subtypes (above 94.1%), with the exception of the HER-2+ mammary carcinomas, in which only 77.8% of the tumors showed sTILs-TIM-3+ (Table 4).

### 3.2. Higher TIM-3 Expression in iTILs and in tTILs Is Associated with a More Benign Tumor Behavior, whereas sTILs-TIM-3+ Are Associated with Aggressive Clinicopathological Features

Recently, the expression of TIM-3+ in TILs with different tumor locations has been studied, showing different associations with the patients’ clinicopathological features. Interestingly, we found a negative correlation between the percentage of iTILs-TIM-3+ and tumor size (*p* = 0.037; Figure 2a) and stage (*p* = 0.035; Figure 2b). In contrast, higher percentages of sTILs-TIM-3+ were significantly associated with a higher malignancy grade (*p* = 0.044; Figure 2c) and with the solid histological subtype (*p* = 0.020; Figure 3a). Finally, a negative correlation was found between the percentage of tTILs-TIM-3+ and the tumor size (*p* = 0.041; Figure 2d), as well as with the metastatic disease and tumor recurrence (*p* = 0.010 and *p* = 0.013; Figure 3b and Figure 3c, respectively). 

### 3.3. Higher TIM-3 Expression in Cancer Cells Is Associated with Unfavorable Clinicopathological Features 

Knowing that TIM-3 in HBC showed pro-tumor effects in cancer cells, its expression was analyzed in feline mammary cancer cells and associations with the clinicopathological features were investigated. Accordingly, higher percentages of carcinoma cells overexpressing TIM-3+ were associated with a positive lymph node status (*p* = 0.025; Figure 3d), with luminal B (*p* = 0.010) and triple-negative basal-like (*p* = 0.038) carcinoma subtypes, contrasting with the luminal A subtype (Figure 3e). Additionally, a positive correlation was found between the percentage of cancer cells-TIM-3+ and tumor malignancy grade (*p* = 0.018; Figure 2e).

### 3.4. Higher Percentages of tTILs-TIM-3+ and sTILs-TIM-3+ Showed Prognostic Value in Triple-Negative FMC Subtype 

The prognostic value of TIM-3 differs among the breast cancer subtypes, being associated with a poor prognosis in the luminal A and luminal B carcinomas, whereas a better prognostic outcome has been reported in triple-negative carcinomas. Taking this into consideration, together with the fact that iTILs-TIM-3+ seem to be a favorable prognostic factor in breast cancer, a prognosis analysis, within the different mammary carcinoma subtypes (LA/LB, HER-2+, and triple-negative), was performed in this study. The median, highest quartile (HQ), and lowest quartile (LQ) were used as cut-point values to stratify the data into high and low expression of TIM-3 in TILs (stromal, intra-tumoral, and total) and tumor cells, to investigate the presence of significant differences in OS and DFS between groups. In triple-negative carcinomas, when the lowest quartile was used a as cut-off point (6% and 5.5% for tTILs-TIM-3+ and sTILs-TIM-3+, respectively), the Kaplan–Meier analysis showed that the presence of tTILs-TIM-3+ and sTILs-TIM-3+ in feline mammary carcinomas is a prognostic factor for DFS and OS. Accordingly, cats with triple-negative mammary carcinoma showing higher percentages of tTILs-TIM-3+ and sTILs-TIM-3+ had longer DFS and OS, respectively, than those with lower tTILs-TIM-3+ (*p* = 0.038; Figure 4a) and lower sTILs-TIM-3+ percentages (*p* = 0.009; Figure 4b).

### 3.5. Cats with Mammary Carcinoma Showed Decreased Serum TIM-3 Levels, with Higher Levels being Associated with a Better Disease-Free Survival

Serum TIM-3 levels were detectable in all serum samples collected from 48 queens with mammary carcinoma and 14 healthy animals (detection limit: 125 pg/mL). For the control and diseased groups, the calculated medians were 1391.8 pg/mL (range 1290.7–1573.8 pg/mL) and 1300.5 pg/mL (range 1213.5–1493.2 pg/mL), respectively. Indeed, the serum TIM-3 levels in cats with mammary carcinoma showed significantly lower concentrations than those in the healthy group (*p* < 0.001; Figure 5).

The receiver operator curve (ROC) analysis of sensitivity versus specificity of the ELISA was performed to determine the best cut-point value for predicting OS and DFS as well as to predict the diagnostic value of the serum TIM-3 levels. The best cut-point value identified was 1322.9 pg/mL, with an area under the curve of 0.8 ± 0.065 (95% CI: 0.674–0.929, *p* = 0.001; sensitivity = 66.7%; specificity = 78.6%; Figure 6a). Additionally, the lowest quartile (LQ = 1264.1 pg/mL), median (MD = 1300.5 pg/mL), and highest quartile (HQ = 1337.7 pg/mL) were used as cut-point values to stratify the data into high and low expression of serum TIM-3 levels. When the highest quartile was used as a cut-point, the Kaplan–Meier analysis revealed that cats with mammary carcinoma showing higher serum TIM-3+ levels had longer DFS than those with lower serum TIM-3+ levels (*p* = 0.033; Figure 6b).

### 3.6. Higher Serum TIM-3 Levels Are Associated with Higher TIM-3 Expression in TILs

The median, the upper quartile (HQ), and the lower quartile (LQ) were used as cut-point values to stratify the cats with mammary carcinoma into high and low expression of TILs-TIM-3+ (stromal, intra-tumoral, and total) and cancer cells-TIM-3+. Groups were then tested to assess their association with serum TIM-3+ levels. When the median was used as a cut-off point (20% and 25.5% for sTILs-TIM-3+ and tTILs-TIM-3+, respectively), results showed an association between cats with mammary carcinoma exhibiting higher percentages of sTILs-TIM-3+ and tTILs-TIM-3+ and higher serum TIM-3+ levels (*p* = 0.023 and *p* = 0.016; Figure 7a and Figure 7b, respectively).

### 3.7. Serum TIM-3 Levels Are Positively Correlated with Serum LAG-3 Levels

Considering our previous results showing that cats with mammary carcinoma have higher circulating levels of several immunomodulatory molecules (IL-6, TNF-α, CTLA-4, PD-1, PD-L1, VEGF-A, VEGFR-1, VEGFR-2, and LAG-3) [33,34,35], the serum TIM-3 levels were evaluated to check for statistical correlations. Interestingly, a significant positive correlation was only found between the serum TIM-3 and serum LAG-3 levels (*p* = 0.008; Figure 8). 

### 3.8. Cats with Mammary Carcinoma Showed No Mutations in the Exon 3 of the TIM-3 Gene 

Taking into consideration the published results on human breast cancer, reporting that the +4259T/G polymorphism (rs1036199) in the exon 3 of the TIM-3 gene is associated with an increased Ki-67 index and with metastatic disease, leading to a worse prognosis and disease progression [36], the nucleotide sequence of the exon 3 was analyzed in 19 tumor samples. After sequence alignment, no SNPs or mutations were found in the exon 3. However, a single-nucleotide substitution was found in the intron 2 (NC_058368.1: g.190379339 G>A), in 5.26% of the studied population (1/19), not present in the human counterpart.

## 4. Discussion

Little is known about the biological role of TIM-3 in HBC. Previous works reported that only 11% of breast cancer patients expressed TIM-3 in iTILs, whereas sTILs-TIM-3+ were detected in 40% of breast tumors [37]. Additionally, iTILs-TIM-3+ are more prone to be detected in triple-negative basal-like breast cancer, with sTILs-TIM-3+ being detected mostly in luminal A and luminal B mammary carcinomas, in contrast with the HER-2+ and the triple-negative mammary carcinomas [38]. Accordingly, the results obtained in this study showed that the iTILs-TIM-3+ are less frequent than sTILs-TIM-3+ in feline mammary carcinomas. Interestingly, in contrast with human reports, in FMC, the percentage of tumors having iTILs-TIM-3+ was higher in the luminal A and luminal B mammary carcinoma subtypes (83.3% and 70.6%, respectively) than in the triple-negative basal-like mammary carcinomas (37.5%). However, the percentage of tumors having sTILs-TIM-3+ was lower in the HER-2+ subtype (77.8%), as reported in human breast cancer patients. Regarding the differential role of TIM-3 expression in TILs with different tumor locations (stromal and intra-tumoral) and in cancer cells, an association between a higher TIM-3+ expression in iTILs and poor clinicopathological features has been reported [37], with similar associations reported for higher frequencies of sTILs-TIM-3+. Accordingly, our investigations also revealed an association between sTILs-TIM-3+ expression and more aggressive clinicopathological features, such as a higher malignancy grade (*p* = 0.044) and solid histological subtype (*p* = 0.020). In contrast with the previous findings reported in human breast cancer, our results suggest an association between the TIM-3 expression in iTILs and in tTILs and a benign tumoral behavior, including lower tumor size (*p* = 0.037; *p* = 0.041) and tumor stage (*p* = 0.035) and a less frequent metastatic disease (*p* = 0.010) and tumor recurrence (*p* = 0.013).

In agreement with previous published data on HBC [29,39], our results showed that higher densities of cancer cells overexpressing TIM-3+ were associated with a positive lymph node status (*p* = 0.025), a higher malignancy grade (*p* = 0.018), and with the triple-negative basal-like subtype (*p* = 0.038). Recent findings in HBC showed that blocking the TIM-3 receptor has remarkable anti-tumor effects, suggesting that TIM-3-targeted therapies are very promising [40]. Our results in feline mammary carcinoma patients indicate that cats might also benefit from novel therapies targeting TIM-3, considering that TIM-3 was highly expressed in both TILs and breast cancer cells. Although the mechanisms which enable breast cancer cells-TIM-3+ to escape the immune response are still poorly described, it has been demonstrated that the TIM-3/galectin-9 pathway is able to protect breast carcinoma cells against cytotoxic T-cell-induced death [41]. 

Furthermore, the soluble form of TIM-3, which inhibits the activation of both NK cells and cytotoxic T lymphocytes [41], was reported to be lower in breast cancer patients compared with healthy controls [41,42], probably because of the binding of the soluble TIM-3 form to TILs in breast cancer patients [41] or due to the binding of the TIM-3 receptor to the galectin-9 ligand expressed on the tumor cell surface, making the tumor-associated TIM-3-galectin-9 complex unlikely to be secreted. In line with these findings, our results clearly demonstrate that serum TIM-3 levels are lower in cats with mammary carcinoma compared with the healthy controls. Within the diseased group, a positive association was found between higher expressions of sTILs-TIM-3+ and tTILs-TIM-3+ and the serum TIM-3 levels. A possible explanation for these results may be related to the TIM-3 inhibitory effect on TILs. In fact, sTILs-TIM-3+, which represent the major component of the tTILs-TIM-3+ population, are mainly composed of an exhausted T-cell phenotype. This T-cell exhaustion results in a deficient immune response. Consequently, a higher expression of TIM-3+ on sTILs may inhibit this population’s capacity to shift to the intra-tumoral compartment and interact with the TIM-3-galectin-9 complex on the tumor cell surface, thus making the TIM-3-galectin-9 complex more prone to proteolytic shedding and secretion.

In prognostic analysis, human breast cancer patients with higher expression of TIM-3 on TILs were associated with a more favorable prognosis in triple-negative carcinomas [43,44]. In line with these findings, our results revealed that tTILs-TIM-3+ is a favorable prognostic factor in triple-negative feline mammary carcinomas. Thus, a higher TIM-3+ expression on tTILs was associated with a longer DFS. Interestingly, a higher sTILs-TIM-3+ expression in triple-negative carcinomas was associated with a greater OS. The Kaplan–Meier analysis also revealed that cats with mammary carcinoma showing higher serum TIM-3+ levels had a more favorable prognostic outcome than those with lower serum TIM-3+ (*p* = 0.033).

Regarding the correlation between the serum TIM-3 levels and the serum levels of other immune checkpoint molecules, recent findings [45] showed that serum LAG-3 levels were lower in HBC compared to healthy controls. Interestingly, in cats with mammary carcinoma, our results revealed a positive correlation between the TIM-3 and the LAG-3 serum levels (*p* = 0.008). LAG-3 could be another highly relevant target in feline cancers, as previous studies in humans reported a positive effect of targeting LAG-3 in association with anti-PD-1 [46].

Finally, in breast cancer, polymorphisms in the TIM-3 gene increase susceptibility and disease progression [16]. Indeed, in the coding region of the TIM-3 gene, the rs1036199 SNP’s (+4259T/G) disrupts the TIM-3 expression, modifying the TIM-3 protein function and increasing the occurrence of several malignant tumors [16]. This polymorphism is known to change the exon 3 (amino acid substitution: Arg140Leu), and thus, the mucin-like domain of the protein [47]. Moreover, a positive association between the TIM-3 +4259T/G SNP and an unfavorable progression and prognosis in invasive breast cancer has been previously reported [36]. Our results showed no mutations in the exon 3 TIM-3 gene, on the 19 tumor samples. However, we were able to find a nucleotide substitution in the intron 2, but just in one animal (1/19, 5.26%). Overall, our results provided a comprehensive overview of the role of TIM-3 in feline mammary carcinoma, but further research is needed to fully understand the role of TIM-3 in cancer.

## 5. Conclusions

In conclusion, our data suggest that the TIM-3 expression in cancer cells and TILs might contribute to predict distinct outcomes in cats with mammary carcinoma. Accordingly, cancer cells-TIM-3+ and sTILs-TIM-3+ are associated with more aggressive clinicopathological features, while total and intra-tumoral TILs expressing TIM-3+ are associated with a more benign behavior. Furthermore, sTILs-TIM-3+ and tTILs-TIM-3+ may represent a novel favorable prognostic factor for cats with triple-negative mammary carcinomas. As described for humans, the results obtained clearly demonstrated that serum TIM-3 levels are lower in cats with mammary carcinoma compared with healthy controls, with higher serum TIM-3 levels being associated with a better clinical outcome and higher TIM-3 expression in total and stromal TILs. In summary, these results support that cats with mammary carcinoma may benefit from the development of novel therapies targeting TIM-3 and reinforce the utility of spontaneous FMC as a model for human breast cancer.

## Figures and Tables

**Figure 1 cancers-15-00384-f001:**
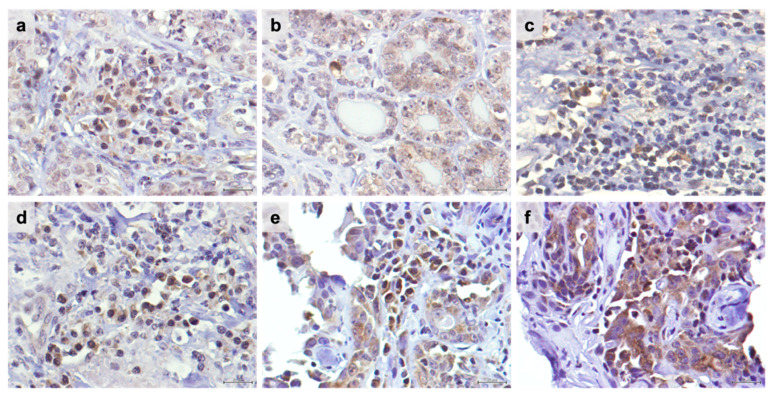
TIM-3 is highly expressed in tumor-infiltrating lymphocytes (TILs) and cancer cells in feline mammary carcinoma. Nuclei were stained by Harris hematoxylin in blue and TIM-3-positive cells in brown. (**a**) Representative TIM-3 immunostaining of intra-tumoral TILs with a moderate staining intensity (2+) in contrast with the weak staining intensity (1+) detected in the surrounding cancer cells (40× objective). (**b**) TIM-3 score intensity 2+ in positive cancer cells, (**c**) in stromal TILs, (**d**) in intra-tumoral TILs, and (**e**) in intra-tumoral TILs and cancer cells (40× objective). (**f**) TIM-3 score intensity of 3+ in cancer cells. Scale bar = 20 μm.

**Figure 2 cancers-15-00384-f002:**
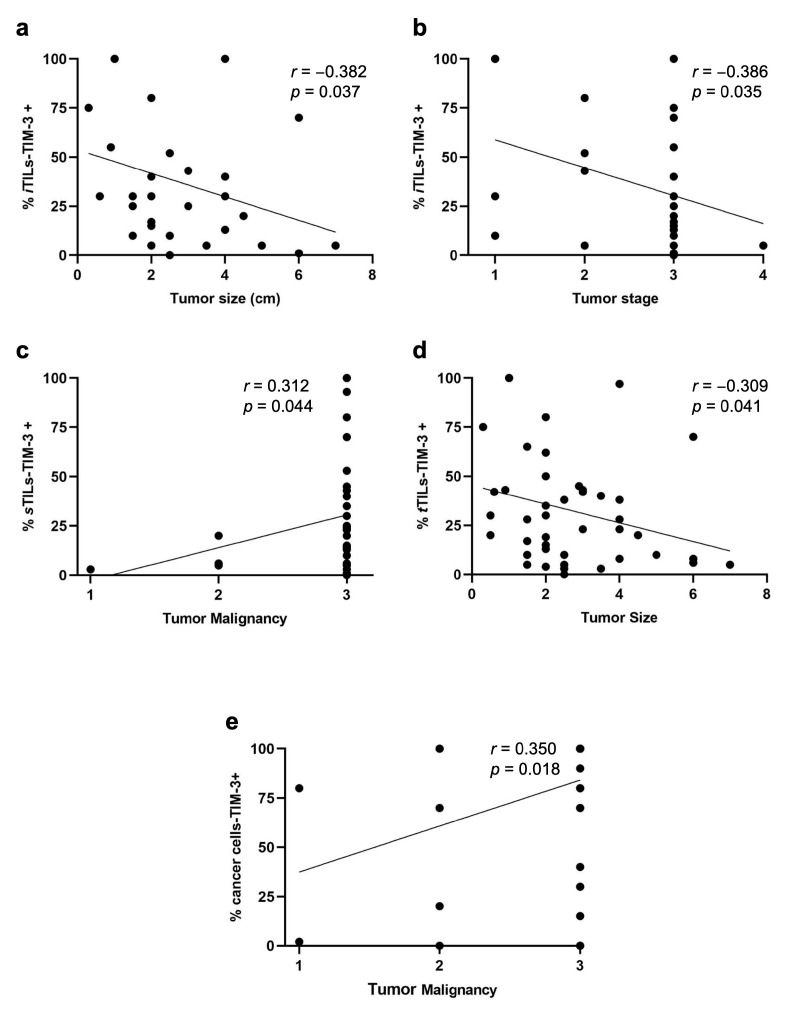
Spearman’s correlations of TILs-TIM-3+ with different tumor locations (intra-tumoral, stromal, and total) and cancer cells-TIM-3+ and their association with clinicopathological features. The percentage of iTILs-TIM-3+ negatively correlates with (**a**) tumor size (*r* = −0.382; *p* = 0.037) and (**b**) tumor stage (*r* = −0.386; *p* = 0.035). (**c**) The percentage of sTILs-TIM-3+ positively correlates with tumor malignancy (*r* = 0.312; *p* = 0.044), and (**d**) the percentage of tTILs-TIM-3+ negatively correlates with the tumor size (*r*= −0.309; *p* = 0.041). (**e**) The percentage of cancer cells-TIM-3+ positively correlates with the tumor malignancy (*r* = 0.350; *p* = 0.018).

**Figure 3 cancers-15-00384-f003:**
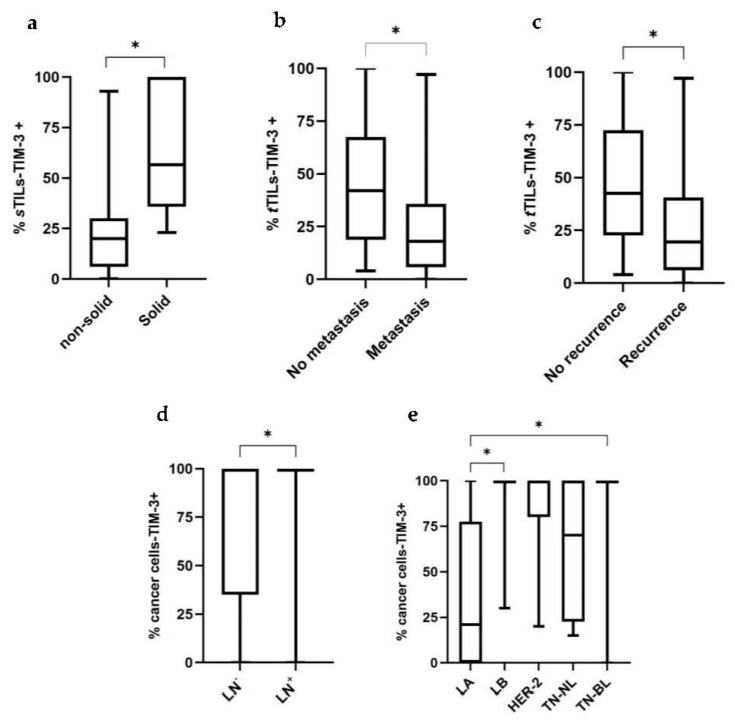
Box plot analysis of stromal and total TILs-TIM-3+ and cancer cells-TIM-3+ and their association with clinicopathological features. (**a**) The percentage of sTILs-TIM-3+ is associated with the solid histological subtype (non-solid, *n* = 39; solid, *n* = 6). (**b**) The percentage of tTILs-TIM-3+ is associated with lower tumor metastasis (no metastasis, *n* = 18; metastasis, *n* = 27) and (**c**) tumor recurrence (no recurrence, *n* = 14; recurrence, *n* = 29). (**d**) The percentage of cancer cells-TIM-3+ is associated with a positive lymph node status (LN) (negative LN status, *n* = 26; positive LN status, *n* = 16) and (**e**) is higher in the LB and TN-BL carcinomas, compared with LA carcinomas (LA, *n* = 6; LB, *n* = 17; HER-2, *n* = 9; TN-NL, *n* = 5; TN-BL, *n* = 8) (* *p* < 0.05).

**Figure 4 cancers-15-00384-f004:**
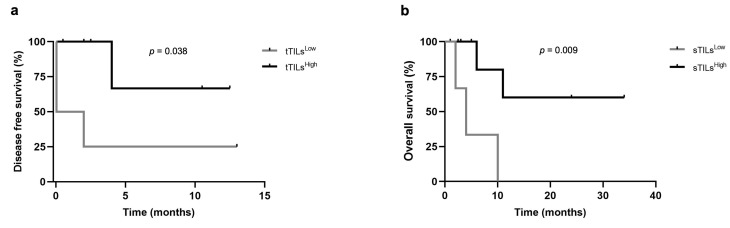
Disease-free survival and overall survival of cats with triple-negative mammary carcinoma according to low or high expression of TIM-3+ on total and stromal TILs. Kaplan–Meier survival curves for association of (**a**) total TIM-3+ TILs with disease-free survival and (**b**) stromal TILs with overall survival.

**Figure 5 cancers-15-00384-f005:**
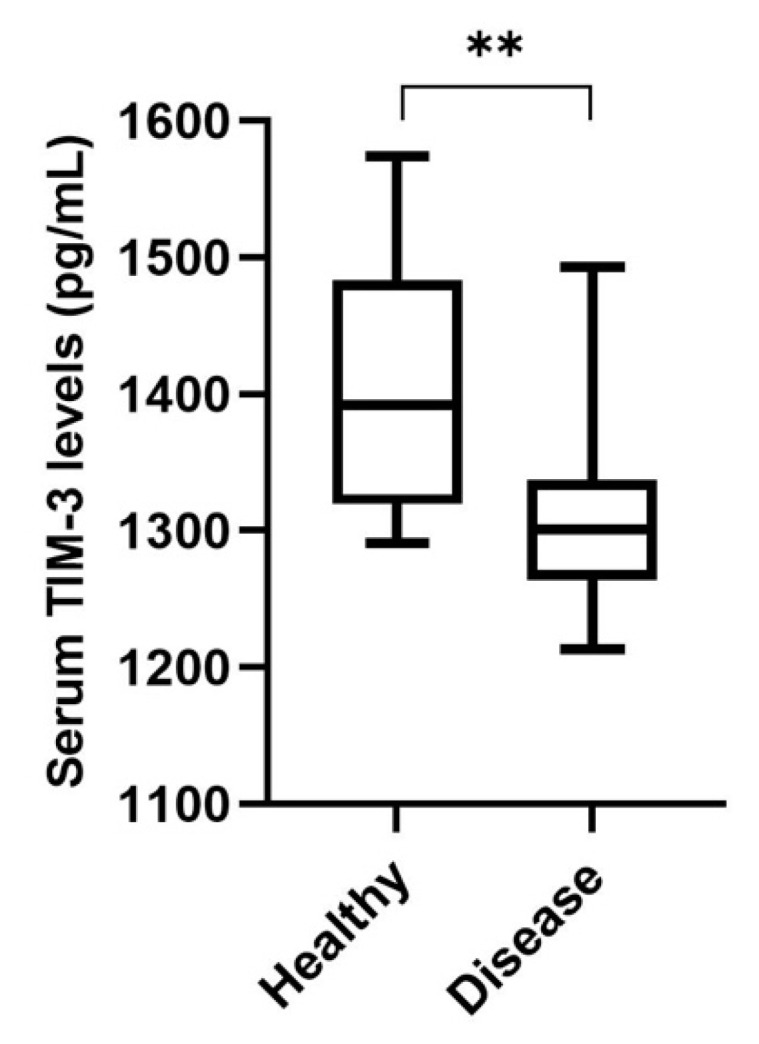
Box plot analysis of serum T-cell immunoglobulin and mucin-domain-containing molecule-3 (TIM-3) levels in healthy cats (*n* = 14) and cats with mammary carcinoma (*n* = 48) (** *p* < 0.01).

**Figure 6 cancers-15-00384-f006:**
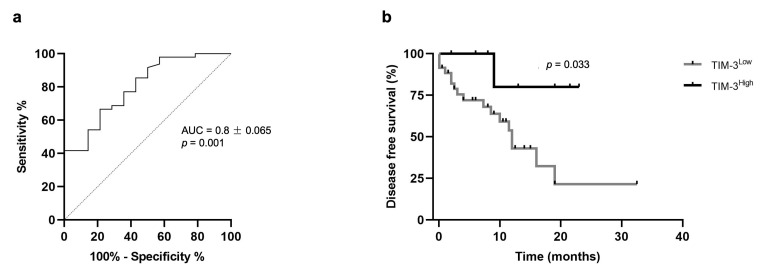
Prognostic value of serum TIM-3 levels. (**a**) Receiver operator curve (ROC) analysis of sensitivity versus specificity for serum TIM-3: AUC 0.8 ± 0.065 (95% CI: 0.674–0.929, *p* = 0.001; sensitivity = 66.7%; specificity = 78.6%). (**b**) Kaplan–Meier survival curve for serum TIM-3: Disease-free survival of cats with mammary carcinoma according to low or high expression of serum TIM-3 levels (*p* = 0.033). The highest quartile was used as a cut-point.

**Figure 7 cancers-15-00384-f007:**
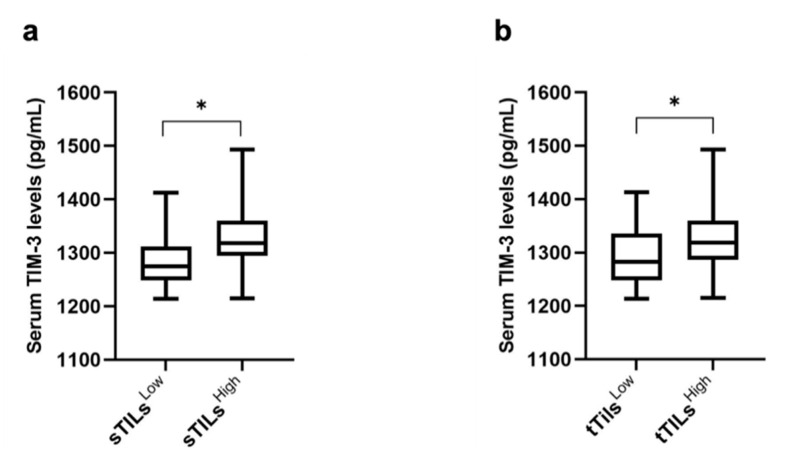
Box plot analysis of serum TIM-3 levels and their association with the TIM-3 expression in stromal and total TILs. (**a**) Serum TIM-3 levels and *s*TILs (sTILs^Low^, *n* = 22; sTILs^High^, *n* = 20). (**b**) Serum TIM-3 levels and *t*TILs (tTILs^Low^, *n* = 22; tTILs^High^, *n* = 23) (* *p* < 0.05).

**Figure 8 cancers-15-00384-f008:**
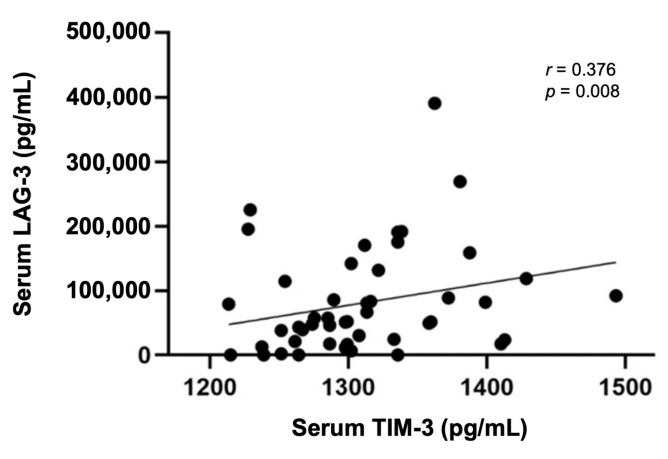
Spearman’s correlation of serum TIM-3 and serum LAG-3 levels (*r* = 0.376; *p* = 0.008).

**Table 1 cancers-15-00384-t001:** Clinicopathological features of 48 cats with mammary carcinoma, enrolled in this study.

Clinicopathological Features	Number of Animals (%)	Clinicopathological Features	Number of Animals (%)
**Age**		**LVI**	
<8 years old	4 (8.3%)	No	41 (85.4%)
8–12 years old	24 (50%)	Yes	7 (14.6%)
>12 years old	31.3 (41.7%)		
**Breed**		**Tumor ulceration**	
Undifferentiated	35 (72.9%)	No	41 (85.4%)
Pure	13 (27.1%)	Yes	7 (14.6%)
**Contraceptive administration**: 6 unknown		**TNM classification**	
No	20 (41.7%)	I	11 (22.9%)
Yes	22 (45.8%)	II	6 (12.5%)
**Spayed**: 1 unknown		III	27 (56.3%)
No	23 (47.9%)	IV	4 (8.3%)
Yes	24 (50.0%)		
**Tumor burden**		**Lymph node status**: 3 unknown	
Single tumor	19 (39.6%)	Negative	29 (60.4%)
Multiple tumors	29 (60.4%)	Positive	16 (33.3%)
**Tumor size**		**ER status**	
<2 cm	15 (31.3%)	Negative	36 (75%)
≥2 cm	33 (68.8%)	Positive	12 (25%)
**HP classification**		**PR status**	
Tubular carcinoma	3 (6.3%)	Negative	20 (41.7)
Tubulopapillary carcinoma	6 (12.5%)	Positive	28 (58.3)
Papillary-cystic carcinoma	6 (12.5%)	**HER-2 status**	
Cribriform carcinoma	19 (39.6%)	Negative	38 (79.2%)
Solid carcinoma	11 (22.9%)	Positive	10 (20.8%)
Mucinous carcinoma	3 (6.3%)		
**Malignancy Grade**		**Ki-67 index**	
1	2 (4.2%)	Low	14 (29.2%)
2	6 (12.5%)	High	34 (70.8%)
3	40 (83.3%)		
**Tumor Necrosis**			
No	11 (22.9%)		
Yes	37 (77.1%)		

HP—histopathologic; LVI—lymphatic vessel invasion; ER—estrogen receptor; PR—progesterone receptor; HER—epidermal growth factor receptor.

**Table 2 cancers-15-00384-t002:** Sequence of the primers used for genomic DNA amplification and sequencing of exon 3 of the feline *TIM-3* gene.

Exon	Forward (5′–3′)	Reverse (5′–3′)
3	ACTAGACTGTGATGACGATGGC	AGGAACATTCACACCTCCACTC

**Table 3 cancers-15-00384-t003:** Percentages (%) and frequencies (*n*) of tumors with TIM-3+ expression in intra-tumoral TILs (iTILs-TIM-3+) among the 45 FMC stratified according to their molecular subtype (luminal A (LA), luminal B (LB), HER-2+, triple-negative normal-like (TN-NL), and triple-negative basal-like (TN-BL)).

Occurrence of TIM-3+ TILs	Tumors with iTILs-TIM-3+ (%)					
	LA	LB	HER-2+	TN-NL	TN-BL	Total
Present	83.3% (*n* = 5)	70.6% (*n* = 12)	66.7% (*n* = 6)	60.0% (*n* = 3)	37.5% (*n* = 3)	64.4% (*n* = 29)
Absent	16.7% (*n* = 1)	29.4% (*n* = 5)	33.3% (*n* = 3)	40.0% (*n* = 2)	62.5% (*n* = 5)	35.6% (*n* = 16)
Total	100% (*n* = 6)	100% (*n* = 17)	100% (*n* = 9)	100% (*n* = 5)	100% (*n* = 8)	100% (*n* = 45)

**Table 4 cancers-15-00384-t004:** Percentages (%) and frequencies (*n*) of tumors with TIM-3+ expression in stromal TILs (sTILs-TIM-3+) among the 45 FMC stratified according to their molecular subtype (luminal A (LA), luminal B (LB), HER-2+, triple-negative normal-like (TN-NL), and triple-negative basal-like (TN-BL)).

Occurrence of TIM-3+ TILs	Tumors with sTILs-TIM-3+ (%)					
	LA	LB	HER-2+	TN-NL	TN-BL	Total
Present	100% (*n* = 6)	94.1% (*n* = 16)	77.8% (*n* = 7)	100% (*n* = 5)	100% (*n* = 8)	93.3% (*n* = 42)
Absent	0% (*n* = 0)	5.9% (*n* = 1)	22.2% (*n* = 2)	0% (*n* = 0)	0% (*n* = 0)	6.7% (*n* = 3)
Total	100% (*n* = 6)	100% (*n* = 17)	100% (*n* = 9)	100% (*n* = 5)	100% (*n* = 8)	100% (*n* = 45)

## Data Availability

The datasets used and analyzed in the current study are available from the corresponding author in response to reasonable requests.

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
