# Peer review of "TIM-3 Is a Potential Immune Checkpoint Target in Cats with Mammary Carcinoma"

_cancers, 2023, doi:10.3390/cancers15020384_

Round 1

Reviewer 1 Report

Valente et al. assessed TIM-3 expression in TILs and cancer cells and discovered that TIM-3 may affect the clinical outcome of cats with FMC, which is consistent with previous findings in HBC. I appreciate the authors’ effort to describe the methods in a detailed manner. The paper is well-written and the results are important, but some areas could be improved. I have a few comments. 

It would also be beneficial to add how findings in this study compare to previously reported studies on human breast cancer and what implications their results have for developing TIM-3 targeted therapies in treating feline mammary carcinoma.

The authors could provide more information on the mechanisms that enable breast cancer cells-TIM-3+ to escape the immune response. This could provide further insight into the role of TIM-3 in mammary carcinoma. Additionally, the authors could have discussed the implications of their findings for the development of novel therapies targeting TIM-3. This could provide a better understanding of the potential benefits of such therapies.

The authors could discuss the implications of their findings for the use of spontaneous FMC as a model for human breast cancer. Overall, the research conclusion provides a comprehensive overview of the role of TIM-3 in feline mammary carcinoma, but further research is needed to fully understand the implications of the findings.

Minor comments:

It would be better to add a color legend/description to the Figure 1 caption.

Figure 3 & 5 – it would help readers to understand the distribution and number of samples better if a scatterplot is added to the boxplot. 

Author Response

Reviewer: 1

Comments to the Author

Valente et al. assessed TIM-3 expression in TILs and cancer cells and discovered that TIM-3 may affect the clinical outcome of cats with FMC, which is consistent with previous findings in HBC. I appreciate the authors’ effort to describe the methods in a detailed manner. The paper is well-written and the results are important, but some areas could be improved. I have a few comments. Dear reviewer, thank you so much for your positive comments on our work.

It would also be beneficial to add how findings in this study compare to previously reported studies on human breast cancer and what implications their results have for developing TIM-3 targeted therapies in treating feline mammary carcinoma. Thank you for your comment. 3 new references of studies in breast cancer were added in lines 61-62.

The authors could provide more information on the mechanisms that enable breast cancer cells-TIM-3+ to escape the immune response. This could provide further insight into the role of TIM-3 in mammary carcinoma. Additionally, the authors could have discussed the implications of their findings for the development of novel therapies targeting TIM-3. This could provide a better understanding of the potential benefits of such therapies. As requested, more information was provided on the mechanisms that enable breast cancer cells-TIM-3+ to escape the immune response (lines 76-89).

The authors could discuss the implications of their findings for the use of spontaneous FMC as a model for human breast cancer. Overall, the research conclusion provides a comprehensive overview of the role of TIM-3 in feline mammary carcinoma, but further research is needed to fully understand the implications of the findings. As requested, discussion was improved (lines 474-476).

Minor comments:

It would be better to add a color legend/description to the Figure 1 caption. Performed (Lines 252- 253).

Figure 3 & 5 – it would help readers to understand the distribution and number of samples better if a scatterplot is added to the boxplot.  As suggested, additional information was added at the captions of figures 3 (lines 302-307) and 5 (line 342), in order to help the readers to understand the distribution and number of samples by groups.

Reviewer 2 Report

This is a well crafted study that emphasizes on the important role of TIM-3 in dictating the clinicopathological features of FMC that would be really interesting to the readers and can open new avenues for immunotherapy in animals. The authors are requested to include few modifications to accomadate improvements: 

Do the authors have details about the treatments the study groups of cats were receiving? If yes, then please include the details? Also, if treatment has any correlation with TIM-3 expression.

 Are there any reports available about ICI therapy in cats or other animals in mammary or some other cancer? If it is available then please try to include in the discussion section explaining the opportunity to develop TIM-3 as a target. 

The author should consider to cite an important review on TIM-3 from Dr Kuchroo's lab (TIM3 comes of age as an inhibitory receptor).

Author Response

Reviewer: 2

Comments to the Author

This is a well crafted study that emphasizes on the important role of TIM-3 in dictating the clinicopathological features of FMC that would be really interesting to the readers and can open new avenues for immunotherapy in animals. Dear reviewer, thank you so much for your positive comments on our work.

The authors are requested to include few modifications to accommodate improvements: 

Do the authors have details about the treatments the study groups of cats were receiving? If yes, then please include the details? Also, if treatment has any correlation with TIM-3 expression. Thank you for bring this point to discussion. Unfortunately, the large majority of animals didn’t receive additional therapies besides the mastectomy, thus statistical studies between chemotherapy and TIM-3 expression were not possible.  

Are there any reports available about ICI therapy in cats or other animals in mammary or some other cancer? If it is available then please try to include in the discussion section explaining the opportunity to develop TIM-3 as a target. Until now, no studies were published on ICI therapy in cat or in other animal species. In the near future, we intend to conduct studies with ICI agents in different cell lines of feline mammary carcinoma.

The author should consider to cite an important review on TIM-3 from Dr Kuchroo's lab (TIM3 comes of age as an inhibitory receptor). Added in line 89.